

# Sigma
## Mobile Tutoring Platform



**Autors**: Wiktor Stasiak · Katarzyna Wysokińska · Jan Mondry · Ishiita Pal

**Supervisor:** Wojciech Thomas

### Abstract

Sigma is a cross-platform tutoring application designed to connect tutors and students, streamlining the tutoring process in a user-friendly environment. The platform serves as a marketplace where tutors can advertise their services and manage schedules, while students can search, compare, and book sessions based on expertise, availability, and ratings. Key features include a session calendar for scheduling, real-time messaging for communication, notifications for updates and reminders, and a tutor rating system to promote quality.

This project aims to simplify traditional tutoring by addressing inefficiencies in finding and managing sessions. Sigma increases accessibility to education by integrating essential tools into a single platform, allowing users to focus on learning rather than logistics. By providing a comprehensive and streamlined solution, Sigma enhances the tutoring experience for both students and tutors.

## 1 INTRODUCTION

The Sigma project addresses the challenge of inefficient and fragmented solutions for tutoring. Existing tools often force users to juggle multiple platforms for scheduling, communication, and material sharing, leading to confusion and wasted time. For tutors, managing sessions, maintaining availability, and advertising their services can be cumbersome. Similarly, students face difficulties in finding suitable tutors and booking sessions efficiently.

Sigma was designed as an all-in-one platform to simplify and streamline the tutoring process. Acting as a marketplace, it connects tutors and students through a user-friendly application that integrates scheduling, messaging, and tutor discovery. By centralizing these functionalities, Sigma eliminates the need for external tools, enabling seamless interaction between users.

Developed using Flutter for cross-platform compatibility and Firebase for backend services, Sigma leverages modern, scalable technologies to ensure reliability and rapid development. These technologies enable a consistent and efficient user experience on Android and iOS devices, catering to the needs of diverse users.

The project aims to achieve the following objectives:

- Develop a cross-platform application using Flutter and Firebase to provide an accessible solution for tutors and students.

- Create an intuitive session calendar for scheduling and managing tutoring sessions.

- Enable students to search for tutors based on subject expertise, availability, and ratings, making comparisons simple and efficient.

- Integrate real-time messaging and notifications to improve communication and ensure users stay updated.

The practical significance of Sigma lies in its ability to enhance operational efficiency, improve tutor visibility, and simplify session management. By addressing these key issues, Sigma empowers students and tutors to focus on learning and teaching rather than logistics.

## 2 TECHNOLOGY STACK

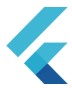 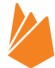 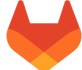 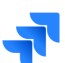 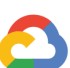 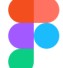 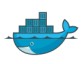

# 3   RELATED WORK

The tutoring industry has seen a surge in digital platforms aimed at connecting students and tutors, such as Wyzant, Preply, and Tutor.com. These platforms serve as marketplaces, allowing users to find and book tutors based on subject expertise and availability. While effective in providing access to tutoring services, these solutions often rely on external tools for session management, communication, and material sharing, which can disrupt workflow and reduce efficiency. Sigma addresses this gap by integrating all essential functionalities into a single, user-friendly application.

Unlike existing platforms, Sigma emphasizes streamlining the entire tutoring process. Features such as an in-app calendar for session scheduling, a real-time messaging system, and push notifications eliminate the need for external tools, enabling seamless interaction between users. Additionally, the platform's focus on tutor ratings and personalized search filters ensures students can quickly find the right match, enhancing the user experience.

From a technological perspective, Sigma leverages Flutter for cross-platform development, ensuring accessibility on both Android and iOS. Firebase is used as the Backend-as-a-Service (BaaS), providing a robust infrastructure for real-time data synchronization, notifications, and user authentication. These choices were driven by the need for scalability, reliability, and rapid development within the constraints of a 10-week project timeline.

The team faced several challenges during development, including:

· Balancing the breadth of features with the limited development time.

· Ensuring seamless integration of core functionalities, such as notifications and messaging.

· Maintaining a user-friendly design while incorporating advanced features like tutor search filters and session management.

Despite these limitations, the chosen technologies and modular design approach allowed the team to focus on delivering an efficient MVP while leaving room for future enhancements. By integrating critical functionalities and prioritizing user experience, Sigma distinguishes itself from competitors and provides a more cohesive solution for tutors and students.

# 4   RESULTS

The Sigma project successfully implemented the key functionalities outlined in the MVP feature set, delivering a robust platform that addresses the inefficiencies of traditional tutoring methods. The following functionalities were completed during the development phase:

· **Session Calendar:** Tutors can create and manage session schedules, while students can view and book sessions seamlessly, streamlining the scheduling process.

· **Tutor Search with Filters:** Students can find tutors based on subject expertise, availability, and ratings, enabling quick and informed decision-making.

· **Real-Time Messaging:** Tutors and students can communicate directly within the platform, improving collaboration and reducing reliance on external tools.

· **Push Notifications:** Users receive timely reminders for upcoming sessions and updates about postponements or cancellations, ensuring engagement and reducing missed sessions.

· **Tutor Ratings:** Students can leave feedback for tutors, promoting transparency and helping others make informed choices.

These functionalities were developed using Flutter for cross-platform compatibility and Firebase for real-time data synchronization and notifications. By integrating these tools, Sigma ensures a seamless and efficient user experience for both tutors and students.

## 4.1   Achievements and Impact

The project achieved the following objectives:

· **Improved Tutor Visibility:** Tutors can effectively advertise their services, expanding their reach to a broader audience of potential students.

- **Simplified Scheduling:** The session calendar allows users to book sessions within minutes, significantly reducing the complexity and time involved in organizing tutoring sessions.

- **Enhanced Communication:** The in-app messaging system ensures consistent and reliable communication between tutors and students.

- **User Engagement:** Notifications and reminders keep users actively engaged, minimizing missed sessions and enhancing the overall experience.

## 4.2 Practical Applications

Sigma's functionalities provide immediate value to both students and tutors:

- **Students:** Quickly find and book tutors that match their specific needs, saving time and improving the overall learning experience.

- **Tutors:** Efficiently manage their schedules, reduce administrative tasks, and focus on delivering quality sessions.

- **Integrated Features:** The platform eliminates the need for external tools, reducing complexity and improving overall user satisfaction.

By achieving the objectives of MVP, Sigma has laid a strong foundation for future iterations. These results position Sigma as a scalable, efficient, and user-friendly tutoring platform capable of addressing the evolving needs of students and tutors. Future iterations will expand on these results, incorporating advanced functionalities such as video conferencing, payment processing, and personalized recommendations.

# 5 CONCLUSIONS

The Sigma project successfully delivered a comprehensive tutoring platform that addresses inefficiencies in traditional tutoring methods. By integrating essential functionalities such as session scheduling, tutor discovery, messaging, and notifications into a single application, Sigma simplifies the tutoring process and enhances the user experience for both tutors and students. The project demonstrates how modern technologies, such as Flutter and Firebase, can be leveraged to create scalable and efficient solutions for educational challenges.

One of the most significant successes of the project was the seamless implementation of the in-app calendar and tutor search features. These functionalities enable students to quickly find and book tutors based on specific criteria, such as subject expertise, availability, and ratings. The intuitive design of these features ensures that users can navigate the platform with minimal effort, saving time and reducing complexity.

Sigma's success lies not only in its technical achievements but also in its potential to create meaningful impact for its users. Tutors benefit from improved visibility and the ability to manage their schedules efficiently, enabling them to focus more on delivering quality education. For students, the platform provides an accessible and reliable way to connect with experienced tutors, improving their educational outcomes and learning experiences.

Additionally, Sigma sets the stage for further advancements in the online tutoring industry. By providing a strong foundation of core features, the platform has the flexibility to adapt and grow with the addition of advanced functionalities. Features such as integrated video conferencing, payment processing, and personalized recommendations have the potential to expand Sigma's reach and effectiveness, making it a versatile tool for both tutors and students.

In summary, the Sigma project showcases how technology can bridge gaps in traditional tutoring systems, making education more accessible, efficient, and engaging. Its accomplishments reflect the hard work and collaboration of the development team, and the platform's success offers a blueprint for similar initiatives aimed at transforming the educational landscape.

# 6 FUTURE DIRECTIONS

Sigma has laid the groundwork for a comprehensive and efficient tutoring platform, but there are numerous opportunities for future development that could enhance its functionality, user engagement, and scalability. Below are key areas for potential improvement and expansion:

- **Advanced Video Conferencing Integration:** While Sigma currently allows tutors and students to share external meeting links, integrating a built-in video conferencing tool or providing seamless integration with platforms like Zoom, Google Meet, or Microsoft Teams would create a more cohesive user experience. A native video solution could include features such as session recording, whiteboard functionality, and screen sharing, tailored specifically for tutoring.

- **Payment Processing System:** Adding a secure, in-app payment system would simplify fee management for tutors and students. Features like payment tracking, invoicing, and flexible payment schedules could enhance the platform's usability and reduce administrative overhead for users.

- **Personalized Recommendations:** Incorporating AI and machine learning algorithms to suggest tutors to students based on their learning history, preferences, and performance metrics would increase user satisfaction. Similarly, tutors could receive recommendations for students based on their expertise and availability.

- **Progress Tracking and Analytics:** A progress tracking module could allow tutors to monitor their students' performance over time, providing insights through metrics like completed topics, test scores, and session attendance. For students, this feature would offer a visual representation of their learning journey, encouraging motivation and continued use of the platform.

- **Resource Sharing and File Attachments:** Enabling tutors and students to share learning materials directly through the platform, such as PDFs, images, and presentations, would enhance session preparation and post-session review. This feature could include version control to ensure users have access to the latest materials.

- **Gamification Features:** Introducing gamification elements, such as badges for achieving learning milestones, session streaks, or rewards for consistent usage, could boost student engagement and encourage retention. Tutors could also benefit from recognition for high ratings or successful sessions.

- **Scalability and Multi-Tenant Architecture:** As the user base grows, implementing a multi-tenant architecture could help manage larger numbers of users, ensuring fast response times and uninterrupted service. This would also allow for regional customization, such as language preferences or currency options for payments.

- **Community Features:** Adding a forum or Q&A section where students and tutors can share tips, resources, or address common learning challenges would foster a sense of community and increase engagement within the platform.

- **Internationalization and Localization:** Expanding Sigma's reach by supporting multiple languages and region-specific features would make it accessible to a global audience. This could include localized pricing, cultural learning adaptations, and multilingual support.

These improvements would not only enhance Sigma's core functionality but also position it as a leader in the online tutoring industry. By focusing on user experience, scalability, and advanced features, Sigma can grow into a comprehensive platform that adapts to the evolving needs of students and tutors worldwide.

## ACKNOWLEDGMENTS

We would like to thank our supervisor, Wojciech Thomas, for his guidance throughout the project.

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
