# OpenReview forum: "Sigma - mobile tutoring platform"
_pwr.edu.pl/Wrocław_University_of_Science_and_Technology/2024/ZPI_Day — Wrocław University of Science and Technology 2024 ZPI Day Submission_

### Official Review · Reviewer_GDnd · 2024-12-05
**Mobile Tutoring Platform**

**Confidence:** 5
**Significance Of Results:** 5
**Overall Quality:** 4

**Compliance With Template:**

5: Very High Quality – The article contains all the required sections, which are written in a very detailed, clear, and error-free manner. The structure is professional and meets expectations, and the content adheres to the highest substantive and formal standards.

**Description Of Results:**

3: Average Quality – The results are described with moderate detail. Some examples or evaluation elements are present but insufficiently developed or incomplete.

**Feedback On Consistency:**

The introduction effectively outlines the inefficiencies and fragmentation in existing tutoring solutions. It highlights specific pain points for both tutors and students, such as managing schedules, communication, and finding suitable tutors. While the problem is identified, the analysis could benefit from more quantitative data or user research findings to substantiate the claims about existing inefficiencies. The related work section briefly mentions competitors like Wyzant and Preply but could delve deeper into their specific shortcomings, providing a stronger case for Sigma's unique value proposition. The presentation of results lacks specific metrics or user feedback that demonstrate the effectiveness or user satisfaction with the implemented features. Including such data would provide a clearer picture of the platform's success. Comparing Sigma’s performance or user engagement metrics against existing platforms could strengthen the argument for its effectiveness and potential impact. While the conclusions recap the achievements, they could more explicitly reference the specific results presented earlier, ensuring a tighter linkage between what was achieved and the overarching conclusions. Acknowledging the current limitations of the MVP and how future developments will address these would provide a more balanced and realistic conclusion.

Enhancing the transitions between problem analysis, results, and conclusions can improve the overall flow, ensuring each section builds seamlessly upon the previous one. Article could benefit from Ensuring that each section not only stands alone but also references relevant parts of other sections can enhance coherence. For example, linking specific results back to the problems they address can reinforce consistency.

**Potential For Development:**

The article clearly highlights numerous opportunities for further development and practical applications of the Sigma project’s outcomes. The proposed directions for growth could not only enhance Sigma's competitiveness in the market but also provide more engaging and effective educational experiences for both tutors and students.

**Project Nature Evaluation:**

The introduction effectively identifies the inefficiencies and fragmentation of existing tutoring solutions, highlighting specific challenges faced by both tutors and students, such as schedule management, communication, and finding suitable instructors. To enhance the analysis, it would be beneficial to include quantitative data or user research findings that could better substantiate the claims about current inefficiencies.

The related works section mentions competitors like Wyzant and Preply; however, a more detailed discussion of their specific shortcomings could better emphasize Sigma's unique value proposition. In the results section, incorporating specific metrics or user feedback to illustrate the effectiveness and user satisfaction with the implemented features would provide a clearer picture of the platform's success. Additionally, a comparison of Sigma's performance and user engagement metrics with existing platforms could strengthen the argument for its efficiency and potential impact.

While it is understandable that conducting extensive research under the constraints of a tight semester schedule can be challenging, but even a limited usability testing effort would add significant value to the project.

**Technical Language Precision:**

4: High Quality – The language is appropriate for a technical report. Terminology is used correctly, and statements are precise, with only minor shortcomings that do not affect the overall clarity.

---

### Official Review · Reviewer_mHvc · 2024-12-08
**IT project prepared correctly.**

**Confidence:** 5
**Significance Of Results:** 5
**Overall Quality:** 5

**Compliance With Template:**

5: Very High Quality – The article contains all the required sections, which are written in a very detailed, clear, and error-free manner. The structure is professional and meets expectations, and the content adheres to the highest substantive and formal standards.

**Description Of Results:**

5: Very High Quality – The results are described in detail, clearly and comprehensively, supported by thorough evaluation, analysis, and convincing usage examples. The description meets the highest substantive standards.

**Feedback On Consistency:**

The analysis of the problem, presentation of results and conclusions are coherent and logical

**Potential For Development:**

The article indicates possibilities for further work or practical application of its results.

**Project Nature Evaluation:**

Both the level of usability, the applied technical methods and technological solutions have the characteristics of engineering work.

**Technical Language Precision:**

5: Very High Quality – The language is entirely appropriate for a technical report. All terms are used correctly and precisely, and the style is professional, clear, and coherent, without any errors or ambiguities.

---

### Official Review · Reviewer_kLYw · 2024-12-08
**Interesting solution, with briefly described results.**

**Confidence:** 5
**Significance Of Results:** 4
**Overall Quality:** 4

**Compliance With Template:**

4: High Quality – The article contains all the required sections, which are well-written and substantively correct, although minor errors or shortcomings may be present. The overall structure is clear and coherent.

**Description Of Results:**

3: Average Quality – The results are described with moderate detail. Some examples or evaluation elements are present but insufficiently developed or incomplete.

**Feedback On Consistency:**

The paper is written in a consistent manner. All parts are included. However, the presentation of results is very brief and general. Actually, there is no slightly deeper information about the architecture of the application. The section "Related work" contains some architectural details of the application that should rather be located in the next "Results" section.
The "Conclusions" section is correct and coherent with the rest of the paper, except for the first sentence: Sigma does NOT address inefficiencies in traditional tutoring methods, as it does not support the tutoring process itself. It is a tool for finding tutors, improving communication, and scheduling meetings, but not to run the tutoring process itself.

**Potential For Development:**

The paper contains a quite wide set of possible extensions of the provided solution. Extending this solution with tools to run a tutoring process would be a huge additional benefit.

**Project Nature Evaluation:**

The team focused on providing the solution to the identified problem: improving communication and scheduling in the tutoring process. The choice of technologies is correct and appropriate for the planned goal. I consider it important, that the team has focused on providing the solution to the existing problem.

**Technical Language Precision:**

5: Very High Quality – The language is entirely appropriate for a technical report. All terms are used correctly and precisely, and the style is professional, clear, and coherent, without any errors or ambiguities.

---

### Decision · Program_Chairs · 2024-12-10

Accept (Poster)